# Respiratory Health among Pesticide Sprayers at Flower Farms in Ethiopia

**DOI:** 10.3390/ijerph19127427

**Published:** 2022-06-17

**Authors:** Meaza Gezu Shentema, Magne Bråtveit, Abera Kumie, Wakgari Deressa, Bente Elisabeth Moen

**Affiliations:** 1Department of Preventive Medicine, School of Public Health, Addis Ababa University, Addis Ababa P.O. Box 9086, Ethiopia; aberakumie2@yahoo.com (A.K.); deressaw@gmail.com (W.D.); 2Research Group for Occupational and Environmental Medicine, Department of Global Public and Primary Care, University of Bergen, 5020 Bergen, Norway; magne.bratveit@uib.no (M.B.); bente.moen@uib.no (B.E.M.); 3Centre for International Health, University of Bergen, 5009 Bergen, Norway

**Keywords:** pesticide sprayers, flower farm, respiratory health, Ethiopia

## Abstract

Background: Pesticide use in Ethiopia has become a common practice in which large-scale flower farms are the main consumers. Workers on flower farms might be exposed to pesticides while spraying or while performing other tasks related to pesticide use and management. It is unclear whether working as a flower farm sprayer is associated with respiratory health problems. Objective: The objective of this study was to compare respiratory symptoms and lung function indices between pesticide sprayers and non-spraying workers. Method: A cross-sectional study was conducted on 15 flower farms, involving all-male sprayers as the pesticide-exposed group and all other male workers as a control group. Data were collected using a standard questionnaire for respiratory symptoms developed by the British Medical Research Council and the American Thoracic Society. Lung function tests were performed to determine forced vital capacity (FVC), forced expiratory volume at one second (FEV_1_), mid 50 expiratory flow, and the ratio of FEV_1_ to FVC. Chi-squared tests and Poisson regression analyses were used to compare respiratory symptoms between the two working groups. General linear regression models were used to compare lung function test indices between spraying and non-spraying working groups. The significance level was set to 0.05. Results: A total of 285 male workers participated (152 sprayers and 133 non-spraying workers). The mean age of the workers was 25 years for sprayers and 24 years for non-sprayers. The proportions of cough, cough with sputum, breathlessness, and wheezing were similar in the two groups, while chest tightness was significantly high in the non-spraying group. Sprayers had significantly higher FVC and FEV1 than the non-spraying group. Conclusions: Respiratory symptoms were not different between the sprayers and non-spraying workers except that the non-spraying workers had increased chest tightness. FVC and FEV1 were significantly higher among sprayers relative to non-sprayers. The results must be interpreted with caution, as the sprayers used respiratory protective equipment, which probably reduced their exposure to the pesticides. Also, the workers were young, and a healthy worker effect might be present among the sprayers.

## 1. Introduction

In developing countries such as Ethiopia, pesticide use is increasing mainly due to the commercialization of large-scale farms [1,2,3]. Ethiopian flower farms are high consumers of pesticides both in terms of diversity and quantity [3]. Pesticides used constitute different chemical compositions intended for use as insecticides, fungicides, and herbicides. According to the World Health Organization (WHO) acute hazard classifications, various pesticides with class I–IV are used [4,5]. A study from six Ethiopian flower farms in 2020 revealed the use of 22–50 different pesticides on each farm [5]. The type of pesticides used in a flower farm shifts from day to day, due to different pests occurring in these large farms [4].

Pesticide sprayers are responsible for the daily spraying of pesticides inside plastic shield greenhouses to enhance the productivity of flower farms [6,7]. The sprayers are a specific group of workers, who are specialized in the spraying tasks. The sprayers use spraying wands for pesticides to spray manually mainly by walking forwards through the cloud of pesticides [4,5]. Inhalation and skin contact are the dominant exposure routes for the work of spraying pesticides [8,9] that might lead to the development of adverse health effects caused by pesticides [6,10]. There are few exposure studies on sprayers from flower farms. The sprayers use many different pesticides, and they may change the types often due to the different pests occurring on the farms. However, previous studies indicate that sprayers are more exposed to pesticides than the other workers on flower farms [11,12].

Exposure to several pesticides is reported to produce irritation in the respiratory system [7,8]. Studies conducted among pesticide spraying workers in the agricultural sector have shown an increased prevalence of respiratory symptoms compared to workers who are not spraying [1,13]. Other studies have indicated that pesticide sprayers develop impaired respiratory function [14,15], and it has been suggested that these workers develop diseases such as asthma, chronic obstructive pulmonary disease and chronic bronchitis [14,15,16]. However, the conclusions from these studies are not clear. In Ethiopia, studies have indicated a high prevalence of respiratory symptoms among flower farm workers, but this is based on respiratory symptom questionnaires and not on objective measurements of lung function [4,16].

Workers in flower farms are also engaged in multiple activities outside the greenhouses, such as handling tasks inside the packinghouse and in the cold storage room. These workers are involved in either trimming flowers or making them ready for packing and transport [5]. These workers do not handle pesticides and rarely enter the greenhouses where pesticides are sprayed [5,17]

In this study, we aimed to assess respiratory symptoms and undertook objective measurements of lung function indices, comparing pesticide sprayers and a control group of non-spraying workers at Ethiopian flower farms. We hypothesized that pesticide exposure in the flower farms may lead to reduced respiratory health among the pesticide sprayers.

## 2. Methods

### 2.1. Study Area and Design

A cross-sectional study was conducted in an area of flower farms within a 50 km radius of the capital city of Ethiopia, Addis Ababa. This area is at a high altitude (2000–2500 m) which favors the production of high-quality rose flowers. The study site is close to Bole International Airport to enhance the exporting of rose flowers. The study areas are characterized by the use of different pesticides including organophosphates, neonicotinoids, pyrethroids and inorganic pesticides. Pesticides of different WHO hazard classes are used, including those with WHO classes I and II [5]. The examinations of the workers took place for 6 weeks, and details on the actual use of pesticides in the flower farms these days were not obtained, as this was not known to the individual workers we examined.

### 2.2. Study Population

Sprayers engaged in pesticide spraying tasks were classified as the exposure group for this study. The sprayers were compared with non-spraying workers. The non-sprayers included workers engaged in packinghouses outside the greenhouse area as a comparative group for the study. These workers were not supposed to enter the greenhouse where sparing activities take place. Non-spraying workers are engaged in trimming flowers for packing and transporting flowers from the packing house to a cold room and vice versa. In the flower farms in Ethiopia, all sprayers are male. Therefore, male workers from packinghouses were chosen as a control group in the present study [5,17].

### 2.3. Sample Size Determination

The sample size for respiratory symptoms was calculated based on a previous study conducted among flower farm workers reporting a 47.8% prevalence of cough among sprayers and 28.6% among other workers [4]. At a power of 90 and 95% confidence interval, the sample size was calculated to be 134 for each group, and when adding a 10% non-response rate the sample size was 147. The sample size for lung function indices was calculated based on the mean difference formula, taking values of forced expiratory volume at one second of sprayers (FEV_1_) [1]. The sample size was calculated to be 160 for a mean (SD) of FEV_1_ of 3.06 (0.59) liters among sprayers and 3.26 (0.45) liters among other workers, when using a 95% confidence interval and a power of 90% and considering a 10% non-response rate. So, we decided to invite 160 participants since it was the larger sample size calculated.

### 2.4. Sampling Procedures

In the study area, there were 31 rose-producing farms. Eighteen of these farms were randomly selected for the study and 15 agreed to participate. In terms of the number of sprayers and other male workers who were invited to participate in the control group, all sprayers (*n* = 155) and male non-spraying workers from the packinghouses (*n* = 141) were invited to participate in the study.

### 2.5. Data Collection

The respiratory symptom data were collected using a standardized questionnaire adopted from British Medical Research Council (BMRC) and the American Thoracic Society (ATS) symptom questionnaire [18,19]. The questionnaire was also previously used for studies on respiratory symptoms among other Ethiopian flower farm workers [4]. The questionnaire was translated into local languages (Amharic and Afan Oromo) to facilitate understanding by all participating workers as needed. A data collector who had an experience in data collection from previous studies collected the data through face-to-face interviews after undergoing two days of training on the tool and questioning technique. The questionnaire comprised background data (age in years, educational status, height in centimeters, weight in kilograms, cooking inside/outside the house and cooking fuel), any previous physician-diagnosed respiratory diseases, behavioral questions (alcohol drinking and smoking status), work-related questions (service duration in months, transfer to the present work from another working section in the same flower farm, hours worked per day) and questions regarding the use of respiratory protective equipment (Yes or no). It also included respiratory symptoms questions regarding cough in the morning, cough during the day or night, cough with sputum in the morning, cough with sputum during the day or night, breathlessness walking (walking up a slight hill, walking with same age group, forcing one to stop walking), chest tightness and wheezing. The operational definition for each respiratory symptom with their respective questions and answers considered for having symptoms are indicated in Table 1.

Lung function data were collected using a Minispir light spirometer, with Winspiro software (Medical International Research. Rome-Italy). All tests were performed in a calm room that was designated for data collection. Each participant was informed as to the nature of the tests by a detailed description of the procedures and techniques of the spirometry test before the test. Forceful exhalation was performed by each participant after a deep breath to the lung capacity. The procedure continued until each participant made three acceptable measurements. The maximum effort made by participants was eight. The maximum forced vital capacity (FVC), forced expiratory volume at one second (FEV_1_), FEV1/FVC, and mid 50 expiratory flow (FEF25-75) were recorded as a final measurement from three acceptable measurements.

### 2.6. Data Management and Statistical Analysis

Data from 25 participants (11 sprayers and 14 non-sprayers) were excluded after checking against the acceptability and repeatability criteria of ATS recommendation for the spirometry test [20]. Descriptive statistics such as frequency and percentage were used to present the data. Continuous variables among sprayers and non-sprayers were compared using independent *t*-tests. Categorical variables were compared using Chi-squared tests. Respiratory symptoms were analyzed by the Poisson regression model using the robust estimate of variance. Lung function indices were analyzed using a multiple linear regression model. Age was categorized into tertiles for statistical ease of analysis as it lacks a linear relationship with lung function indices. In our analysis, adjustments were performed for hours worked per day, educational status, and cooking place (inside or outside their house) because these factors were significantly different between sprayers and non-spraying workers. We also adjusted for current smoking and age as they are known predictors of respiratory symptoms [21,22]. Service months correlated with age (r = 0.37, *p*-value < 0.01), and therefore the analyses were made both with and without service years included. Statistical package for social sciences (SPSS) version 25 was used for analysis and a *p*-value of <0.05 was considered statistically significant.

### 2.7. Ethical Consideration

Ethical approval was obtained from the institutional review board of the College of Health Science, Addis Ababa University. A support letter was written to each farm from the school of public health at Addis Ababa University. Each farm administrator was asked for permission after which each selected worker was requested for written consent after reading the information sheet prepared for the study. The information sheet contained information on the purpose of the study, confidentiality, and voluntary participation.

## 3. Results

Overall, 285 workers (152 spraying and 133 non-spraying workers) participated with a 96% response rate. The median service duration was 24 months (18 for sprayers and 24 for non-sprayers. Sprayers worked fewer hours per day when compared to non-sprayers (*p* < 0.05). Non-sprayers had a higher education when compared to sprayers: 40.6 vs. 27.0% with secondary and above education level. More non-spraying workers (65.4%) reported cooking inside the main house than the sprayers (51%). There was no difference in mean age, height, service duration and body mass index (BMI) among the spraying and non-spraying workers (Table 2). The use of respiratory protective equipment (RPE) while working was reported by 145 (96%) sprayers. Four sprayers reported that the reason for not using RPE was that this was not provided at the workplace.

### 3.1. Respiratory Symptoms

Breathlessness and chest tightness were the most frequently reported respiratory symptoms. The two study groups had no significant difference in cough, cough with sputum, wheezing and breathlessness. Sprayers reported a significantly lower prevalence of chest tightness when compared to non-sprayers workers (Table 3). This was consistent in a Poisson regression model analysis, while adjusting for hours worked per day, educational status, and cooking place (inside and outside the main house). There was no change in the estimates when including service months in the regression models.

### 3.2. Spirometry Indices

The sprayers had a significantly higher value of FVC and FEV1 when compared to non-sprayers (Table 4). No significant differences between the groups were observed for either FEV_1_/FVC or FEF_25–75%_ (Table 4). This was consistent when adjusting for age group, educational status, cooking place, and service durations.

## 4. Discussion

There was no significant difference in reported symptoms between sprayers and non-spraying flower farm workers except for the symptom of chest tightness, which was lower among sprayers. Sprayers had a significantly higher FVC and FEV_1._ than non-sprayers. The findings indicate that working as a sprayer is not related to reduced respiratory health at the flower farms, and our hypothesis was not confirmed.

The prevalence of cough, cough with sputum (phlegm) and breathlessness in our study population was comparable with the prevalence of respiratory symptoms among pesticide sprayers from a previous study in Ethiopia, which for instance found the prevalence of cough to be 12.6, compared to 10.1% in the present study [2]. However, the prevalence of each of the respiratory symptoms in our present study was lower when compared to another study conducted among three flower farms in Ethiopia [4]. This could be due to the sample size difference and selection bias, as the latter study was conducted among only 23 sprayers involving only two flower farms, while we examined 285 workers in 15 flower farms. The prevalence of coughing and difficulty in breathing was found to be lower in our study when compared to what was reported among cut flower farmers in Japan [11]. This might be due to the greater use of respiratory protective equipment among sprayers in our study (96%), compared to the Japanese study that reported 29 (42%) who did not use protection [11].

The lung function indices FVC and FEV_1_ for sprayers (4.42 and 3.59 L) and non-sprayers (4.21 and 3.50 L) in our present study were lower when compared to a healthy Ethiopian non-smoking population of men aged 38–47 years (4.57 and 3.79 L) [23]. We also compared findings with a control group of some previous studies in other industries. The FVC and FEV_1_ of non-sprayers and sprayers in our present study are smaller than what was reported among soft drink factory workers (4.41 and 3.63 L) [24].

The findings in our present study did not confirm our hypothesis. This does not necessarily mean that pesticide exposure is not related to adverse respiratory health effects. The population in the present study had a mean age of 26 and had worked for less than 3 years on the flower farm. This short exposure time makes it less likely that the workers could have developed adverse health effects caused by their work.

A healthy workers’ effect, defined as removing sick workers from the work, might play a major role in this study. According to the Ethiopian Horticulture Producers and Exporters Association Exporters Association (EHPEA) code of conduct, flower farm owners are mandated to monitor the health of farm workers. Monitoring the health of these workers may lead to the dismissal of workers who develop health problems. As a consequence, this group of workers might represent a very selected, healthy population. The bronze level, a minimal certification of the EHPEA code of conduct, requires personnel who handle pesticides to have shower facilities and be trained on the risks of pesticide handling and personal protective equipment use [25]. Studies have also confirmed that sprayers of flower farms in Ethiopia are more trained in pesticide handling and have more knowledge than other workers [2].

Another type of healthy workers’ effect is the selection of healthy workers into the work, which can be the reason for the observed low respiratory health problems in sprayers when compared to non-sprayers. During the hiring process, employers will be prone to favor selecting the healthiest population for a strenuous job such as spraying and avoid those with any kind of weakness. Employers also tend to choose non-smokers and non-drinkers of alcohol, as well as those active in sports. Working as sprayers requires physical strength and good skills, and a healthy worker is preferable for employment [8]. In hazardous occupational settings, it’s common that sensitive workers either quit their job early or move to lower exposure sections. Workers from tasks such as spraying might leave the work section because the work is too demanding, and move to a less demanding work section. Among the non-sprayers in our study, 22 (16.5%) had worked in other work sections. This could happen through the administrative decision of the company or workers’ own decisions [8,26]. Sprayers being young and having a short duration of employment might have been accompanied by high turnovers that may have contributed to the better respiratory health among sprayers.

The respiratory protective equipment used by sprayers examined in the present study might have decreased the pesticide exposure and reduced potential negative health effects. Unfortunately, we did not obtain details on the protective masks used by the sprayers. Previous studies on flower farms in Ethiopia indicated the efficient use of personal protective equipment while workers are on duty. Protective pieces of equipment used by sprayers in flower farms include half-face masks, gloves, boots and overall clothes [4,5,27]. The use of personal protective equipment might have decreased pesticide exposure and any potential adverse health effects. A study on the cholinesterase levels of workers had also shown an equivalent level of abnormal values between sprayers and non-sprayers, which might suggest the protection by PPE use among spraying workers [5].

A strength of this study is that we collected objective measurements of lung function besides questionnaire-based interviews of respiratory symptoms. This might help us in managing the possible information bias. Some of the workers might be afraid to reveal symptoms to the researchers during interviews, while other workers might exaggerate their symptoms. Spirometry is an objective measure and is not much influenced by information bias from the workers participating. Another strength of this study is the high response rate among the workers. We compared workers within the farm, both pesticide spraying and non-spraying workers. This was done to reduce the effect of different social statuses, which would have been a problem if we had tried to establish a population-based control group.

We did not monitor pesticide exposure in this study, as previous studies have indicated the use of diversified pesticides in flower farms making it difficult for monitoring [2,3,5]. Three flower farms did not accept our request to participate in our study. We did not have information on the health status of workers working there but this could have introduced selection bias in our study. Farms with a lack of safety systems might not want to participate in such studies, and a negative health effect might have been underestimated in the material.

## 5. Conclusions

Pesticide sprayers in Ethiopian flower farms had less frequency of respiratory symptoms and had better lung function parameters than a group of non-spraying workers. The use of respiratory protective equipment among the sprayers might have reduced the potential adverse effect of pesticides on respiratory health. Further studies are needed to measure the pesticide exposure at the flower farms and to examine the use and efficiency of the protective equipment among sprayers. Long-term studies among flower farm workers are warranted.

## Figures and Tables

**Table 1 ijerph-19-07427-t001:** Operational definitions of different respiratory symptoms.

Symptoms	Questions
Cough *Responding “Yes” to any of the questions*	Do you usually cough first thing in the morning?
	2. Do you usually cough during the day or at night?
	3. Do you usually cough as much as 4–6 times a day for 4 or more days in a week?
	4. Do you usually cough on most of the days for as much as 3 consecutive months or more in a year?
Cough with sputum *Responding “Yes” to any of the questions*	Do you usually cough with sputum first thing in the morning?
	2. Do you usually cough with sputum during the day or at night?
	3. Do you usually cough with sputum as much as 4–6 times a day for 4 or more days in a week?
	4. Do you usually cough with sputum on most of the days for as much as 3 consecutive months or more in a year?
Breathlessness *Responding “Yes” to any of the questions*	Are you troubled by shortness of breath when hurrying on level ground or walking up a slight hill? Do you get shortness of breath walking with other people of your age on level ground? Do you have to stop for a breath walking at your own pace on level ground?
Chest tightness *Responding “Yes” to the question*	Do you usually experience chest tightness while at work or just after work?
Wheezing *Responding “Yes” to the question*	Have you had attacks of wheezing in your chest at any time?

**Table 2 ijerph-19-07427-t002:** Background characteristics of spraying and non-spraying flower farm workers in Ethiopia (*n* = 285).

Variable		Sprayers (*n* = 152)	Non-Sprayers (*n* = 133)	*p*-Value
Age in years; Median (range)		25 (18–60)	24 (18–58)	0.34 ^1^
Height in cm; AM (SD)		171 (6.6)	172 (6.4)	0.62 ^1^
Weight in kg; AM (SD)		58 (6.9)	58 (7.3)	0.86 ^1^
Body Mass Index; AM (SD)		19.9 (1.99)	19.8 (2.04)	0.53 ^1^
Hours worked per day; AM (SD)		5.7 (1.57)	7.9 (0.46)	<0.01 ^1^
Service months; AM (SD)		31.4 (31.1)	37.5 (42.8)	0.17 ^1^
Education	No formal education (%)	23 (15.1)	17 (12.8)	0.05 ^2^
	1^0^ educations (%)	88 (57.9)	62(46.6)	
	2^0^ and above (%)	41 (27.0)	54 (40.6)	
Previous respiratory disease	Yes (%)	4 (2.6)	10 (7.5)	0.057 ^2^
Cooking place inside the main house	Yes (%)	73 (51.4)	85 (65.4)	0.02 ^2^
Transferred from other work section	Yes (%)	17 (11.2)	22 (16.5)	0.19 ^2^
Use of respiratory protective equipment	Yes (%)	145 (96.0)		
Smoking	Yes, no (%)	10 (6.6)	15 (11.3)	0.16 ^2^

AM = arithmetic mean SD = standard deviation, ^1^ = Students *t*-test, ^2^ = Chi-square test.

**Table 3 ijerph-19-07427-t003:** Respiratory symptoms among spraying and non-spraying flower farm workers in Ethiopia (*n* = 285).

Variable	Sprayers (*n* = 152) No (%)	Non-Sprayers (*n* = 133) No (%)	*p*-Value ^1^	Adjusted Prevalence Ratio (95% CI) ^2^
Cough	19 (12.5)	10 (7.5)	0.39	1.4 (0.66–2.90)
Cough with sputum	10 (6.6)	6 (4.5)	0.63	1.25 (0.47–2.29)
Breathlessness	32 (21.1)	18 (13.5)	0.25	1.34 (0.81–2.22)
Chest tightness	15 (9.9)	24 (18.1)	0.02 *	0.49 (0.27–0.89)
Wheezing	8 (5.3)	8 (6.0)	0.75	0.85 (0.29–2.41)

^1^*p*-value from Chi-square test; ^2^ Poisson regression adjusting for age, height, educational status,-working hours per day and cooking inside the main house. * significance is <0.05.

**Table 4 ijerph-19-07427-t004:** Spirometry indices among spraying and non-spraying flower farm workers in Ethiopia (*n* = 256).

Spirometry Indices	Sprayers (*n* = 138)	Non-Sprayers (*n* = 118)	*p*-Value ^1^	Beta (95% CI of Beta)
	AM (SD)			
FVC (l)	4.42 (0.59)	4.21 (0.64)	<0.01 *	0.25 (0.13–0.38)
FEV_1_ (l)	3.59 (0.50)	3.50 (0.58)	0.01 *	0.13 (0.02–0.23)
FEV_1_/FVC	82.0 (5.83)	82.2 (9.67)	0.15	−1.3 (−2.7–0.17)
FEF_25–75%_	3.79 (0.97)	3.85 (1.20)	0.91	−0.04 (−0.29–0.21)

AM = arithmetic means SD = standard deviation; ^1^
*p*-value from multiple linear regression adjusted for age group, height, educational status, current smoking, cooking place (inside and outside the main house) and service month CI = Confidence interval. * significance is <0.05.

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
