# Peer review of "Respiratory Health among Pesticide Sprayers at Flower Farms in Ethiopia"

_ijerph, 2022, doi:10.3390/ijerph19127427_

Round 1

Reviewer 1 Report

In this paper, the respiratory symptoms and lung function parameters were tested to evaluate the effects of pesticide exposure on pesticide sprayers and non-spraying workers. The sample number is large enough and the investigation scheme is reasonable, thereby obtaining the detailed and reliable data. Nevertheless, the result that pesticide sprayers in Ethiopian flower farms had less frequency of respiratory symptoms and had better lung function parameters than a group of non-spraying workers defies common sense. The author did not give a reasonable explanation and sufficient discussion. It is suggested that the author might fully read more literatures based on the background information of the subjects obtained from the investigation, and discuss the possibility that the applicators may receive more stimulation to improve their respiratory and lung functions, so as to make the results of the paper more credible. Additionally, the manuscript needs to be polished by experts with good English skills.

Author Response

Dear Reviewer, thank you for your comments and suggestions on our manuscript. Because of your comments and suggestion our manuscript has improved a lot. Here we have uploaded a point-by-point response document to show our corrections based on your comments and suggestions.

Sinscerly

Meaza Gezu Shentema

Reviewer 2 Report

The paper need to revision from authors. In particular the authors have to follow the instruction of the paper some point need to revision: The number of the reference in the text have to in square parenthesis; The tables need to re-write according to the instruction of the paper; The references have to write according to the instruction of the paper:

In additon in the paper the authors should give information on the type of pesticides that was used for example if the study was performend after spray fungicide, insectides and so on. This information is need to better understand the study and its results.

The authors should give more information on the type of pesticide involved in the study for example if the operator used fungcide or insecticide during spraying. If the application was performed by hand or by manual pump or motor pump. During the application if they used gloves or other protective wear or only respiratory musk.  In addition it should be give information on the application rate of the products.

The authors should distingued between operator that apply the product and worker that enter after spraying and how many time after spraying the worker enter.

These information are important to have specific information and better understand the results obtained.

The results should be detailed and can be described in graphic form to better comprehension.

Author Response

Dear Reviewer, Thank you very much for your comments and suggestions. We have made changes according to your comments and suggestion which helped us in improving the manuscript well. In the following table, we have tried to show a point-by-point response to each of your comments.

Sinscerly

Meaza Gezu Shentema

Reviewer 3 Report

Good study. Authors did not specify the type of respiratory protective equipment used by the sprayers but that would have been helpful to the reader. As a variable, the type of respirator used could help explain the negative findings in the sprayers. Otherwise nice study with objectives lung function measurements.

Some editorial suggestions:

Line 35, suggest "pesticide use is increasing" not "increases"

Line 40, suggest "who specialize" not specialized

Line 82, suggest (6). not . (6).

Line 124, suggest "maximal mid-expiratory flow rate"

Line 150, suggest "College of Health Sciences"

Line 152, suggest "School of Public Health"

Line 169, suggest not capitalizing "S"

Line 215, suggest 10.1%, not "10,1%" 

Author Response

Dear Reviewer 

Thanks a lot for the constructive feedback. We have tried to address your comments and suggestions point-by-point and uploaded it here.

Sinscerly

Meaza Gezu Shentema

Round 2

Reviewer 2 Report

Thank you for the reply.  Paper acceptbale.